# Intestinal *Myo*-Inositol Metabolism and Metabolic Effects of *Myo*-Inositol Utilizing *Anaerostipes rhamnosivorans* in Mice

**DOI:** 10.3390/ijms26199340

**Published:** 2025-09-24

**Authors:** Aldo Grefhorst, Antonella S. Kleemann, Stefan Havik, Antonio Dario Troise, Sabrina De Pascale, Andrea Scaloni, Max Nieuwdorp, Thi Phuong Nam Bui

**Affiliations:** 1Department of Internal and (Experimental) Vascular Medicine, Amsterdam University Medical Centers, 1105 AZ Amsterdam, The Netherlands; 2Proteomics, Metabolomics & Mass Spectrometry Laboratory, ISPAAM, National Research Council, 80055 Portici, Italysabrina.depascale@cnr.it (S.D.P.);

**Keywords:** *Anaerostipes rhamnosivorans*, stable isotopes, *myo*-inositol, metabolism, short chain fatty acids, portal vein

## Abstract

The gut microbiome is strongly implicated in the development of obesity and type 2 diabetes mellitus (T2DM). A recent study demonstrated that 6-week oral supplementation of *Anaerostipes rhamnosivorans* (ARHAM) combined with the prebiotic *myo*-inositol (MI) reduced fasting glucose levels in mice. In the present study, we investigated the effects of a 13-week ARHAM-MI supplementation in high-fat diet-fed mice and examined the metabolic fate of MI, including its microbial conversion into short-chain fatty acids (SCFAs), using ^13^C-MI and stable isotope tracers in the cecum, portal vein, and peripheral blood. The results showed that the ARHAM-MI group gained less weight than the MI-only and placebo groups. Analysis of intestinal mRNA and stable isotope tracing revealed that MI is primarily absorbed in the upper gastrointestinal tract, whereas microbial conversion to SCFAs predominantly occurs in the cecum and is enhanced by ARHAM. ARHAM-MI mice also showed increased cecal *Gpr43* mRNA expression, indicating enhanced SCFA-mediated signaling. Notably, SCFAs derived from MI displayed distinct distribution patterns: ^13^C-butyrate was detected exclusively in the cecum, ^13^C-propionate was present in the cecum and portal vein, whereas ^13^C-acetate was the only SCFA detected in peripheral blood. Collectively, ARHAM-MI co-supplementation confers modest metabolic benefits in high-fat diet-fed mice, underscoring the need to optimize the dosage and administration frequency of ARHAM-MI to enhance its therapeutic efficacy.

## 1. Introduction

It is projected that one in three adults will be affected by obesity and type 2 diabetes mellitus (T2DM) in 2050 [1], primarily driven by an unhealthy lifestyle that includes increased consumption of processed food and physical inactivity. Emerging evidence implicates the gut microbiome as a key player in pathogenesis of both obesity and T2DM, with affected individuals showing distinct alterations in microbial composition and a loss in essential microbial functions compared to healthy counterparts [2,3]. Therefore, understanding the causal relationship between the gut microbiome and metabolic diseases is crucial to unlock the potential of the former as a therapeutic target.

The gut microbiome influences host metabolism by, amongst others, fermenting dietary fibers into the short-chain fatty acids (SCFAs) butyrate, propionate, and acetate [4]. These SCFAs act as signaling molecules implicated in modulation of obesity and T2DM [5]. Butyrate and propionate activate G-protein-coupled receptors (GPCRs), thereby promoting the secretion of incretin hormones such as glucagon-like peptide-1 (GLP-1) and peptide YY (PYY), which enhance glucose metabolism and insulin secretion [6,7]. Butyrate additionally activates intestinal gluconeogenesis whereas propionate serves as a direct substrate for this process, both contributing to glucose homeostasis via the gut–brain neural circuit [8]. SCFAs also support the integrity of the intestinal epithelial barrier, a critical defense against chronic inflammation associated with metabolic diseases [9]. While butyrate is predominantly utilized as an energy source by colonocytes [9], propionate and acetate are partially metabolized by enterocytes, with the unmetabolized fractions entering the portal vein and reaching the liver and other peripheral tissues [10]. Research has shown that individuals with obesity and T2DM exhibit significantly lower levels of butyrate and propionate-producing bacteria [11,12]. Consistently, oral and intraperitoneal supplementation of butyrate and propionate has been shown to attenuate weight gain and improve glucose tolerance in mice [13,14]. However, studies in humans have demonstrated limited effects [15], suggesting that effective treatment strategies may require the use of butyrate and propionate-producing bacteria.

Indeed, evidence from preclinical and clinical studies suggests that oral supplementation of butyrate-producing bacteria may enhance host metabolism and protect against metabolic diseases. For instance, a clinical trial demonstrated that supplementation with the butyrate-producing bacterium *Anaerobutyricum soehngenii* improved insulin sensitivity in individuals with metabolic syndrome [16] and enhanced glucose management in T2DM patients [17]. Consistent results were reported in murine models, where *A. soehngenii* administration led to favorable outcomes in body weight regulation and hepatic physiology [18,19]. These findings highlight the therapeutic potential of butyrate-producing bacteria in metabolic disease management, while it remains to be established whether propionate-producing species confer similar benefits.

Bacteria of the genus *Anaerostipes* are common members of the core human gut microbiome [20]. To date, only four *Anaerostipes* species have been isolated and characterized from stool samples of mammals [21]. These species are well known for their capabilities of converting either monosaccharides, such as glucose, or lactate and acetate to butyrate [22,23]. Further characterization of isolates within these species has revealed their ability to adapt to host dietary intake [24]. For example, some *Anaerostipes* strains can ferment inositol or possess at least the genetic potential to do so. Some *Anaerostipes* strains have been shown to produce propionate by fermenting inositol [24] or dietary phytate [25]. In the latter case, metabolic cross-feeding with the phytate-degrading *Mitsuokella* was required. Altogether, this highly versatile metabolic capacity of *Anaerostipes* species underscores their therapeutic potential in targeting metabolic diseases.

Efforts to identify a microbiome signature associated with metabolic risks have revealed that inositol fermentation by *Anaerostipes* species is reversely associated with metabolic biomarkers in humans [26]. This aligns with earlier studies that show that *myo*-inositol (MI) exhibits anti-inflammatory properties [27], reduces fasting plasma insulin and glucose concentrations in women with gestational diabetes [28], and improves insulin sensitivity and glucose tolerance in T2DM patients when co-supplemented with its isomer D-chiro-inositol [29]. Furthermore, oral MI supplementation can reduce plasma cholesterol and triglyceride concentrations in postmenopausal women [30] and downregulate hepatic mRNA expression of lipogenic genes in rodents [31]. In previous work, we demonstrated that co-supplementation of MI and *Anaerostipes rhamnosivorans* (ARHAM) for 6 weeks reduced fasting glucose levels in mice, an effect associated with increased cecal propionate concentrations [24]. Building on these findings, the present study explores the impact of prolonged ARHAM-mediated MI fermentation in high-fat-diet (HFD)-fed obese mice. Additionally, we aimed to trace the metabolic fate of administered MI by delivering ^13^C-MI and tracking the ^13^C signal in MI and SCFAs across the cecum, portal vein, and peripheral blood. We found that ARHAM-MI-treated mice exhibited lower weight gain relative to their food intake and modestly improved glucose handling compared to mice receiving MI alone. These effects may result from enhanced fermentation of MI to SCFAs by ARHAM, which activates Gpr43 signaling in the cecum. Future studies are warranted to determine whether optimizing the dosage and frequency of ARHAM-MI can further improve its therapeutic efficacy.

## 2. Results

### 2.1. ARHAM Colonizes the Intestinal Tract with Modest Metabolic Effects via Intestinal MI Fermentation

To evaluate the therapeutic potential of ARHAM via MI fermentation, male mice with HFD-induced obesity were administered 200 µL ARHAM suspension (10^9^ CFU/dose) combined with MI (0.1 mg/g body weight), or MI alone (0.1 mg/g body weight), or PBS solution as placebo every other day by oral gavage for 13 weeks. Body weight and food intake were recorded weekly. Metabolic assessments included an intraperitoneal insulin tolerance test (ITT) and an oral glucose tolerance test (OGTT) two weeks and one week before termination, respectively. To investigate the conversion of MI into SCFAs, and the systemic distribution and hepatic first-pass clearance of MI and its derived SCFAs, the mice received an oral challenge of [^13^C_6_]-MI either 2 or 4 h prior to termination. While body weight gain did not differ between mice receiving MI alone and those receiving placebo, mice treated with the ARHAM-MI combination exhibited significantly lower body weight gain compared to both the MI-only or placebo groups (Figure 1A). Interestingly, the ARHAM-MI group also showed increased food consumption (Figure 1B) and a higher weekly estimated food intake relative to body weight (Figure 1C). These findings suggest that the ARHAM-MI treatment conferred protection against diet-induced obesity in this model. Microbial analysis of colonic content confirmed successful colonization of ARHAM in the ARHAM-treated mice, while it was undetectable in the other two groups (Figure 1D).

Upon the ITT, the ARHAM-MI-treated mice exhibited comparable glucose concentrations to the MI-only and the placebo group following insulin injection (Figure 2A). The area under the curve (AUC) of the glucose vs. time curve for the ITT showed a tendency to be reduced in the ARHAM-MI-treated mice compared to the MI group (487 ± 26 vs. 548 ± 27, ARHAM-MI vs. MI, *p* = 0.084). The OGTT results showed that ARHAM-MI treatment resulted in improved glucose handling compared to the placebo-treated mice, particularly at 60 min (*p* = 0.058, ARHAM-MI vs. Placebo) and at 90 min (*p* = 0.061, ARHAM-MI vs. Placebo) post-glucose challenge, although the AUC did not differ (Figure 2B). Notably, treatment with ARHAM-MI or MI alone did not alter the weights of the liver, the gonadal and the inguinal white adipose tissue (gWAT and iWAT) depots, and the intrascapular brown adipose tissue (BAT) depot (Appendix A).

### 2.2. Metabolism of Produced SCFAs and MI in the Cecum, Portal Vein and Systemic Circulation

SCFAs are recognized as key mediators of many beneficial metabolic effects attributed to the gut microbiome [4]. To investigate the in vivo conversion of MI into SCFAs, mice were administered an oral challenge of ^13^C_6_-MI. Cecal contents, portal vein blood and systemic blood were collected 2 or 4 h post-administration for targeted liquid chromatography mass spectrometry analysis.

No difference in the cecal ^13^C-MI concentrations were observed after 2 h, but after 4 h the cecal ^13^C-MI concentrations were significantly lower in the ARHAM–MI group than those in the control group, though not different from the MI-only group (Figure 3A). These results suggest that the endogenous murine microbiome also possesses the capacity to ferment MI, and addition of the exogenous MI-fermenting strain ARHAM appeared to further promote microbial MI utilization. In contrast, no differences were observed between the groups in terms of ^13^C-MI concentrations in portal and systemic blood, suggesting that host MI absorption and metabolism were comparable across all groups. Of note, ^13^C-MI was detectable in the portal vein (~0.2 mM) and peripheral blood (~0.04 mM) of all groups but with a significant reduction after 4 h compared to 2 h post-gavage, suggesting efficient absorption by the enterocytes and metabolism, i.e., hepatic clearance of MI.

The ^13^C analysis revealed active microbial fermentation of ^13^C-MI into SCFAs in the cecum (Figure 3B–D). Significant increases in concentrations in the cecum were observed between 2 and 4 h post-gavage for ^13^C-acetate and ^13^C-butyrate. Of interest, the ARHAM-MI combination resulted in higher ^13^C-acetate in the portal vein 2 h after administration. While ^13^C-butyrate was exclusively detected in the cecal content, ^13^C-propionate was present in the cecum and the portal vein, and ^13^C-acetate was detected in cecum, portal vein, and the systemic circulation. These distribution patterns suggest distinct absorption and metabolic kinetics for individual SCFAs produced from MI fermentation. The concentration of ^13^C-propionate in the portal blood was only a fraction of that in the cecum and remained undetectable in peripheral circulation, implicating that propionate is primarily metabolized by the intestinal epithelium, with only a minor fraction used by the liver. In contrast, ^13^C-acetate concentration was comparable between portal and circulating blood, indicating its concurrent utilization by both hepatocytes and peripheral tissues.

### 2.3. Effects of ARHAM via Inositol Fermentation in the Intestine

To investigate microbial utilization versus host absorption of MI along the gastrointestinal tract, we assessed the mRNA expression of the genes encoding the sodium ion coupled inositol transporter Slc5a3 [32] and the SCFA receptor Gpr43 [33] in the duodenum, ileum, jejunum and cecum (Figure 4). The results showed that *Slc5a3* mRNA expression was significantly higher in the duodenum than the ileum, suggesting predominant MI transport activity in the upper gastrointestinal tract (Figure 4A). In contrast, *Gpr43* mRNA expression did not differ significantly along the intestinal tract (Figure 4C). Interestingly, *Gpr43* mRNA expression was significantly induced in the ARHAM-MI group compared to the placebo group (*p* = 0.039, Figure 4D), indicating that MI fermentation by ARHAM may stimulate Gpr43-mediated signaling in the cecum following ARHAM-MI treatment.

### 2.4. Addition of ARHAM to MI Affects the mRNA Expression of Multiple Lipid-Related Genes in the Inguinal WAT Depot

Next, we determined the effect of the MI treatment with or without ARHAM on the mRNA expression of genes encoding metabolic enzymes as well as *Gpr43* and *Slc5a3* in various metabolic relevant organs. Compared to the placebo-treated mice, the ARHAM-MI combination significantly reduced hepatic *Gpr43* mRNA expression and a tendency towards lower *Slc5a3* mRNA expression (Appendix A). The mRNA expression of other tested genes was not affected by the ARHAM-MI combination. In skeletal muscle, none of the measured genes were affected by the treatments (Appendix A).

In the inguinal white adipose tissue (iWAT), significant effects were seen of the ARHAM-MI combination compared to MI alone (Figure 5). The mRNA expression of the genes encoding the insulin receptor substrate 1 (*Irs1*), lipoprotein lipase (*Lpl*), the fatty acid transporter Cd36, the crucial mediator of triglyceride synthesis diacylglycerol *O*-acyltransferase-2 (*Dgat2*), and the lipid droplet protein cell death-inducing DFFA-like effector-c (*Cidec*) was reduced by the AHAM-MI treatment. Moreover, mRNA expression of the genes encoding fatty acid oxidation (*Cpt1a*) and its transcriptional activators (*Ppara* and *Pgc1a*) tended to be induced by the ARHAM-MI combination in comparison to MI alone. *Pgc1a* mRNA expression is a marker of browning of WAT, hence we also explored the mRNA expression of uncoupling protein-1 (*Ucp1*), which was undetectable in iWAT in all three treatment groups.

## 3. Discussion

Multiple preclinical and clinical studies have demonstrated that modulation of the gut microbiome holds therapeutic potential for the management of obesity and T2DM [3]. Cohort studies have further identified that a structural variation deletion spanning MI catabolic genes in *Anaerostipes hadrus* is associated with metabolic biomarkers such as BMI, and fasting plasma insulin and triglyceride levels [26]. Similar associations were observed in a Swedish (pre)diabetes cohort (*n* = 1011) [24]. This structural variation was predicted to encode proteins involved in the conversion of MI into butyrate, potentially contributing to its metabolic benefits. However, our previous experimental data revealed that many *Anaerostipes* spp. metabolize MI primarily to propionate rather than butyrate [24]. In the same study, we also demonstrated that treating mice with a combination of ARHAM and MI for 6 weeks reduced fasting blood glucose concentrations. To further explore the therapeutic potential of MI fermentation by ARHAM and the metabolic fate of MI and MI-derived SCFAs, the present study evaluated whether the ARHAM-MI combination confers more pronounced metabolic benefits through propionate production after 13 weeks of treatment in mice with diet-induced obesity. Our results revealed that ARHAM-MI-treated mice exhibited reduced weight gain relative to food intake and modest improvements in glucose handling compared to mice receiving MI alone.

Using ^13^C-labeled MI, we traced the metabolic fate of MI and MI-derived metabolites in vivo. The analysis of the ^13^C-MI and the associated ^13^C-SCFAs across different compartments demonstrated that orally administered MI undergoes both host absorption and microbial fermentation, potentially explaining the more pronounced beneficial metabolic effects observed when MI was administered together with ARHAM. Consistent with previous finding [34], the mRNA expression of the MI transporter Slc5a3 in the duodenum suggests that MI is predominantly absorbed in the upper gastro-intestinal tract. In contrast, the microbial fermentation of MI most likely occurs in the cecum, which harbors the majority of gut microbes. Indeed, ^13^C-SCFAs profiling in the cecum confirmed active microbial conversion of MI to SCFAs by both the endogenous murine microbiota and the administered ARHAM strain. The ability to quantify ^13^C-MI derived SCFAs in portal vein as well as peripheral blood provided novel insights into their distinct metabolic fates. Notably, ^13^C- butyrate was undetectable in both blood samples, consistent with its intensive utilization by colonocytes as an energy source [9]. ^13^C-propionate was detected at low concentrations in portal blood only, likely reflecting its role in intestinal gluconeogenesis [8]. Importantly, ARHAM-MI-treated mice exhibited a significant induction of *Gpr43* mRNA expression in the cecum, indicating SCFA-mediated signaling by ARHAM. ^13^C-acetate was the only ^13^C-MI-derived SCFA detected in both portal and systemic circulation, consistent with its broad utilization in the intestinal epithelium, liver and peripheral tissues. Together, our stable isotope-based approach, combined with compartment-specific sampling provided key insights into MI absorption, microbial fermentation, and SCFA metabolism. These findings suggest that targeted delivery of MI to the gut microbiota may enhance metabolic outcomes through increased SCFA production.

The ARHAM-MI combination influenced both body weight regulation and glucose handling, which may be partially mediated by enhanced Gpr43-dependent signaling originating from the cecum. Gpr43, a fatty acid receptor highly expressed in the intestine, has been implicated in the etiology and pathogenesis of obesity and T2DM [35]. One possible explanation for the observed phenotype is increased thermogenesis in the adipose tissues in the ARHAM-MI group compared with the MI-only and placebo groups. However, no significant reductions in fat mass or upregulation of thermogenic genes such as *Pgc1a* [36] were detected, suggesting that the metabolic benefits are unlikely to be attributable to enhanced adipocyte thermogenesis.

To further investigate the underlying mechanisms, we examined the mRNA expression of key metabolism-associated genes in the liver and iWAT. No significant differences in hepatic gene expression were observed among treatment groups, despite previous reports of reduced hepatic lipogenic gene expression following MI supplementation in rats fed a high-sucrose diet [31]. This discrepancy may reflect differences in dietary models, as our study employed a high-fat diet, which is generally less potent in inducing hepatic lipogenesis than refined carbohydrates—rich diets that more closely recapitulate features of human fatty liver disease [37]. Additionally, differences in the mode of MI administration could contribute to the contrasting results; whereas earlier studies incorporated MI directly into the diet, we administered MI by oral gavage at a relatively low dose, three times per week.

In the iWAT depot, mRNA expression of lipoprotein lipase (*Lpl*) and the fatty acid transporter (*Cd36*) was reduced in the ARHAM-MI group compared with the MI-only group, suggesting a potential decrease in triglycerides-derived fatty acid uptake [38]. In contrast, previous studies in 3T3-L1 adipocytes reported that exposure to SCFAs increased *Lpl* mRNA expression [39] and enhanced lipolysis [40]. Thus, our findings indicate that the observed changes in iWAT gene expression in the ARHAM-MI group are unlikely to be mediated by SCFA activity. This interpretation is further supported by our ^13^C-SCFAs tracing analysis, which demonstrated that ^13^C-propionate and ^13^C-butyrate did not escape the hepatic first-pass metabolism, while peripheral blood levels of ^13^C-acetate remained unchanged across treatment groups. Finally, it should be noted that mRNA expression does not necessarily reflect functional protein activity, and therefore it may not fully account for the physiological outcomes observed.

The current studies mainly focused on the role of the SCFAs in the effects of the ARHAM-MI co-treatment. It should however be noted that other microbial-derived metabolites as well as direct effects of the microbiome and its metabolites on the epithelial barrier function might also have had an impact. Future research might be directed to study the role of these factors in controlling metabolic health.

In summary, our study demonstrates that oral supplementation with ARHAM in combination with MI under the present experimental conditions confers modest metabolic benefits, including attenuated weight gain and improved glucose homeostasis after 13 weeks. These effects are likely mediated by enhanced microbial fermentation of MI and activation of Gpr43 signaling in the cecum. Notably, our approach employed low-dose, intermittent oral administration of MI, in contrast to most studies that use continuous dietary supplementation. Future investigations are needed to determine whether optimized dosing regimens and administration frequency of both MI and ARHAM can further improve these metabolic outcomes.

## 4. Materials and Methods

### 4.1. Mouse Study

The mouse study was conducted according to the guidelines of and after approval from the AMC Animal Welfare Committee under protocol number DLV23-16970-1-01, in accordance with the ARRIVE guidelines.

Male C57BL/6J mice from Charles River, ordered at 4 weeks of age, were housed per 3 in a cage and acclimatized for one week. After this week, the mice were fed a high fat diet (HFD; 60% kcal from fat; ResearchDiets D12492i; composition in Appendix A) and received three times a week gavage with PBSA as placebo, 0.1 mg/g MI (Merck, Darmstadt, Germany) in PBS, or 0.1 mg/g MI with 10^9^ ARHAM cells in 15% trehalose. Body weight and food intake were measured weekly. After 12 and 13 weeks, the mice were subjected to an insulin tolerance test (ITT) and an oral glucose tolerance test (OGTT), respectively, of which the details are described below. One week after the OGTT, the mice were sacrificed. For this, the mice were fasted for 5 h in the morning after which they received a gavage with 0.1 mg ^13^C_6_-MI per gram bodyweight. For the mice in the ARHAM-MI group, this gavage also contained 10^9^ ARHAM cells in 15% trehalose. Two or 4 h after ^13^C_6_-MI gavage, the mice were brought under isoflurane anesthesia after which a blood aliquot from the portal vein was collected and the mice were killed by cardiac puncture. The portal vein blood and the cardiac blood samples were stored immediately at 4 °C and various tissues were dissected, weighed, and snap-frozen in liquid nitrogen. The distinct interscapular brown adipose tissue (BAT) depot and the gonadal and inguinal white adipose tissue (WAT) depots were collected as reported by Bagchi and MacDougald [41]. Small intestines were removed and divided into three equal sections. Approximately 5 mm in the middle part was collected and used to determine mRNA expression in the jejunum.

### 4.2. Insulin Tolerance Testing (ITT)

The mice were fasted for 4–6 h after which the blood glucose concentration was measured in a small blood drop from the tail with a handheld ContourXT blood glucose meter. Next, 0.75 IU/kg human insulin (NovoRapid) dissolved in a solution of 1% *w*/*v* BSA in PBS was administered intraperitoneally. Glucose levels were recorded at various time points after the insulin administration. The area under the curve (AUC) of the blood glucose vs. time curve was calculated using GraphPad Prism 10.

### 4.3. Oral Glucose Tolerance Test (OGTT)

The mice were fasted for 4–6 h after which the blood glucose concentration was measured in a small blood drop from the tail with a handheld ContourXT blood glucose meter. Next, 1 g/kg glucose dissolved in demi water was given by gavage. Glucose levels were recorded at various time points after the glucose administration. The area under the curve (AUC) of the blood glucose vs. time curve was calculated using GraphPad Prism 10.

### 4.4. Colonic ARHAM Content

The content of the colon was mixed with STAR buffer (Roche Diagnostics, Germany), homogenized by a bead beater (Bertin Precyllus 24, Bertin, Montigny-le-Bretonneux, France), incubated for 15 min at 95 °C in a heat shaker at 1000× *g*, centrifuged 14,000× *g* at 4 °C after which the supernatant was collected. This procedure was repeated 3 times with the empty tubes to ensure full recovery. The supernatant was mixed with nuclease-free water and placed into the Maxwell RSC 48 instrument (Promega, Madison, WI, USA) to extract DNA. The DNA concentration was measured with the Qubit dsDNA BR assay kit (ThermoFisher Scientific, Waltham, MA, USA) with the Qubit fluorometer (ThermoFisher Scientific, Waltham, MA, USA) according to the manufacturer’s instructions. A quantitative RT-PCR was performed using SensiFAST SYBRgreen (Bioline, London, UK) and primers for 16S and an ARHAM-specific gene (see Appendix A for sequences) with a CFX384 Real-Time PCR System (Bio-Rad, Hercules, CA, USA) to determine the concentration of ARHAM.

### 4.5. Targeted Liquid Chromatography Tandem Mass Spectrometry of Mouse Cecum, Portal Vein and Plasma Samples

Reference compounds, [^13^C_6_]-*myo*-inositol, [^13^C_2_]-acetate, [^13^C_3_]-propionate and [^13^C_4_]-butyrate were obtained from Merck (Darmstadt, Germany). Water, acetonitrile, methanol and formic acid were of mass spectrometry grade, while all the other reagents were of analytical grade. For sample preparation, caecum contents were suspended in 75% methanol then homogenized by sonication (15 min, in ice). Upon centrifugation (18,000× *g* for 10 min, 4 °C), supernatants were used for analysis. Two separate analytical procedures included a 3-nitrophenylhydrazine (3-NPH) derivatization for ^13^C-labeled SCFAs and a HILIC separation for [^13^C_6_]*myo*-inositol; in both cases, a Vanquish core liquid chromatographic system was interfaced to an Expolris 120 high resolution mass spectrometer (Thermo Fisher Scientific, Bremen, Germany).

#### 4.5.1. Quantitation of [^13^C_6_]-*Myo*-Inositol

For [^13^C_6_]-*myo*-inositol quantitation, 10 µL of plasma and portal vein or 100 µL of caecum supernatants were diluted in ice cold methanol (ratio 1:3 *v*/*v*). Suspensions were centrifuged at 18,000× *g* for 10 min, at 4 °C, and 30 µL were dried under vacuum in a centrifugal evaporator (Savant, Thermo Fisher Scientific). Dried samples were resuspended in a solution consisting of 50% *v*/*v* acetonitrile in water and diluted in 80% *v*/*v* acetonitrile according to the linearity range of the calibration curves. [^13^C_6_]-*myo*-inositol was separated at 35 °C through a zwitterionic column (Atlantis Premier BEH, Z-HILIC, 100 *×* 2.1, 1.7 µm, Waters, Milford, MA, USA) with the following gradient of solvent B (minutes/%B): (0/5), (1/5), (10/50), (13/50). Mobile phases consisted of 0.1% *v*/*v* formic acid in acetonitrile (solvent A) and 0.1% *v*/*v* formic acid in water (solvent B); the flow rate was 0.2 mL/min. For targeted quantification of the precursor ion of [^13^C_6_]-*myo*-inositol (^13^C_6_H_12_O_6_, [M-H]^−^ at *m*/*z* 185.0762, with a mass accuracy below 2 ppm, to avoid mass overlap with the natural isotope distribution of non-labeled MI), the heated electrospray (H-ESI) interface parameters were the following: static spray voltage −2.9 kV, ion transfer tube and vaporizer temperature were both at 280 °C; sheath gas flow and auxiliary gas flow were 30 and 15 arbitrary units, respectively. The analyzer resolution was set at 60,000 (FWHM at *m*/*z* 200), with an automatic maximum injection time mode and a standard all ions gain control mode (AGC target). Mild trapping of precursor ion and a chromatographic peak width of 6 s were used. The linear calibration curve was built in the range 0.01–10 mM and concentration values reported in mM were measured through the standard addition technique by using quality controls of caecum and plasma as blank samples.

#### 4.5.2. Quantitation of ^13^C-Labeled SCFAs

For the analysis of fully ^13^C-labeled SCFAs, namely [^13^C_2_]-acetate, [^13^C_3_]-propionate and [^13^C_4_]-butyrate in caecum, portal vein and mice plasma, samples were quantified according to the procedure detailed by Rampanelli et al. [42], including the same derivatization and chromatographic separation protocols with minor modifications. For the ^13^C_2_-, ^13^C_3_- and ^13^C_4_-hydrazone derivative quantitation, chromatographic stream was interfaced to an Exploris 120 working in negative ion mode scanning the ion in the *m*/*z* range 100–400; resolution was set at 60,000 (FWHM at *m*/*z* 200), capillary temperature was 300 °C, vaporizer temperature 280 °C, while sheath and auxiliary gases were set at 45 and 10 arbitrary units, respectively. Analytes profile data in full MS mode were collected using TraceFinder (v. 5.1, Thermo Fisher Scientific). Linear calibration curves were built in the range 1–1000 µM and the concentrationss reported in mM were measured through the standard addition technique by using the following precursor ions for [^13^C_2_]-acetate (^13^C_2_C_6_H_9_N_3_O_3_, [M-H]^−^ at *m*/*z* 196.0638), [^13^C_3_]-propionate (^13^C_3_C_6_H_11_N_3_O_3_, [M-H]^−^ at *m*/*z* 211.0828) and [^13^C_4_]-butyrate (^13^C_4_C_6_H_13_N_3_O_3_, [M-H]^−^ at *m*/*z* 226.1018), all with a mass accuracy below 2 ppm to control overlap with natural isotope contributes of non-labeled SCFAs.

### 4.6. RNA Extraction and Quantitative RT-PCR Analyses

Total RNA was isolated from frozen tissues using TriPure Isolation Reagent (Roche Applied Sciences, Almere, The Netherlands) according to manufacturer’s instructions. Reverse transcription was performed using a cDNA synthesis kit (SensiFAST cDNA synthesis kit, Bioline) according to the manufacturer’s instructions. Quantitative RT-PCR was performed using SensiFAST SYBRgreen (Bioline) with a CFX384 Real-Time PCR System (Bio-Rad). Primer sequences are listed in Appendix A. The expression of each gene was reported in arbitrary units after normalization to the average expression level of the reference genes using the 2^−ΔΔCt^ method.

### 4.7. Statistical Analyses

Figures were rendered in GraphPad Prism 10. Outliers were identified using the ROUT method with Q = 1% and excluded if appropriate. Two-way ANOVA or 1-way ANOVA tests were performed in which the data were compared to ARHAM-MI. *p* < 0.05 was considered significant.

## Figures and Tables

**Figure 1 ijms-26-09340-f001:**
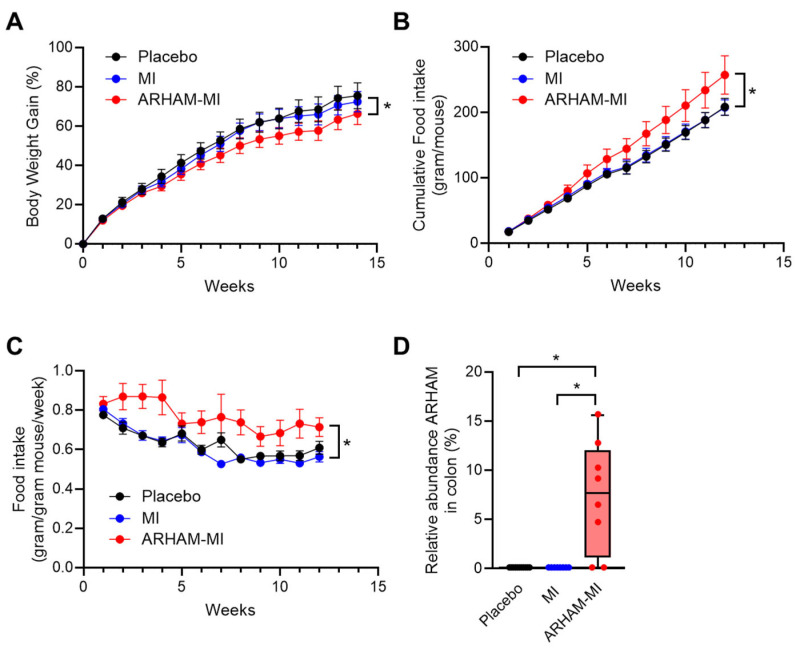
Effects of 12-week treatment with ARHAM-MI or MI alone on body weight, food intake and ARHAM abundance of HFD mice. (**A**) Body weight over time; (**B**) Cumulative food intake per mouse over time; (**C**) Food intake per mouse body weight over time; and (**D**) Relative abundance of ARHAM in the colonic content. Data shown as averages ± SEM or boxplots with 25th–75th percentiles and whiskers following Tukey’s method; *n* = 9–12 mice per group; a, *p* < 0.05 ARHAM-MI vs. Placebo in 2-way ANOVA (panels A–C); b, *p* < 0.05 ARHAM-MI vs. MI in 2-way ANOVA; *, *p* < 0.05 in 1-way ANOVA compared to ARHAM-MI (panel D).

**Figure 2 ijms-26-09340-f002:**
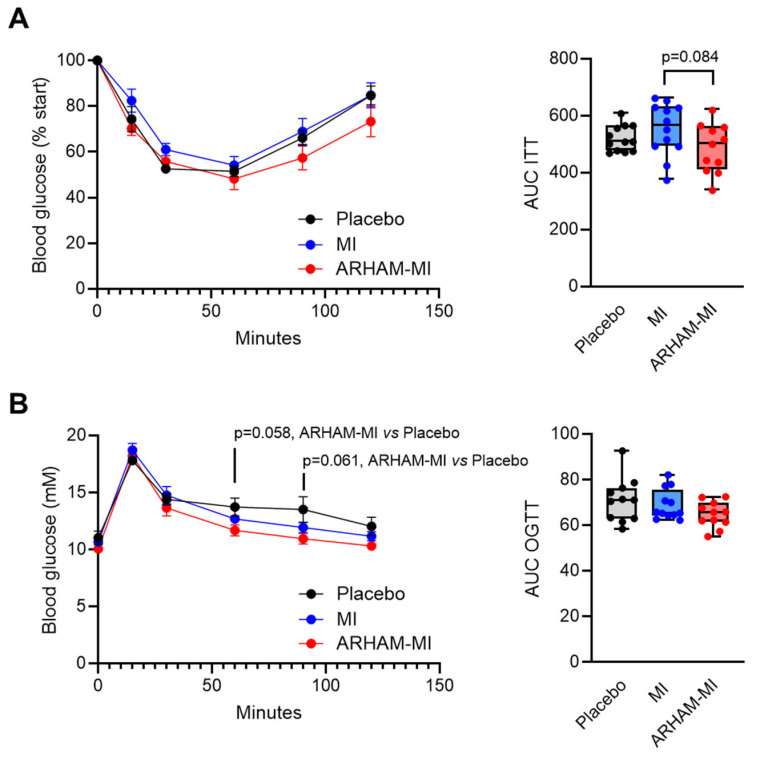
Effects of 13-week treatment with ARHAM-MI or MI alone on insulin tolerance and oral glucose tolerance of HFD mice. (**A**) Blood glucose response after intraperitoneal administration of 0.75 IU/kg insulin, shown as the percentage of the basal glucose level at the time of administration, and area under the curve of this graph; (**B**) Blood glucose response after oral administration of 1 g/kg glucose, and area under the curve of this graph. Data shown as averages ± SEM or boxplots with 25th–75th percentiles and whiskers following Tukey’s method; *n* = 11–12 mice per group; *p*-values are from 1-way ANOVA compared to ARHAM-MI.

**Figure 3 ijms-26-09340-f003:**
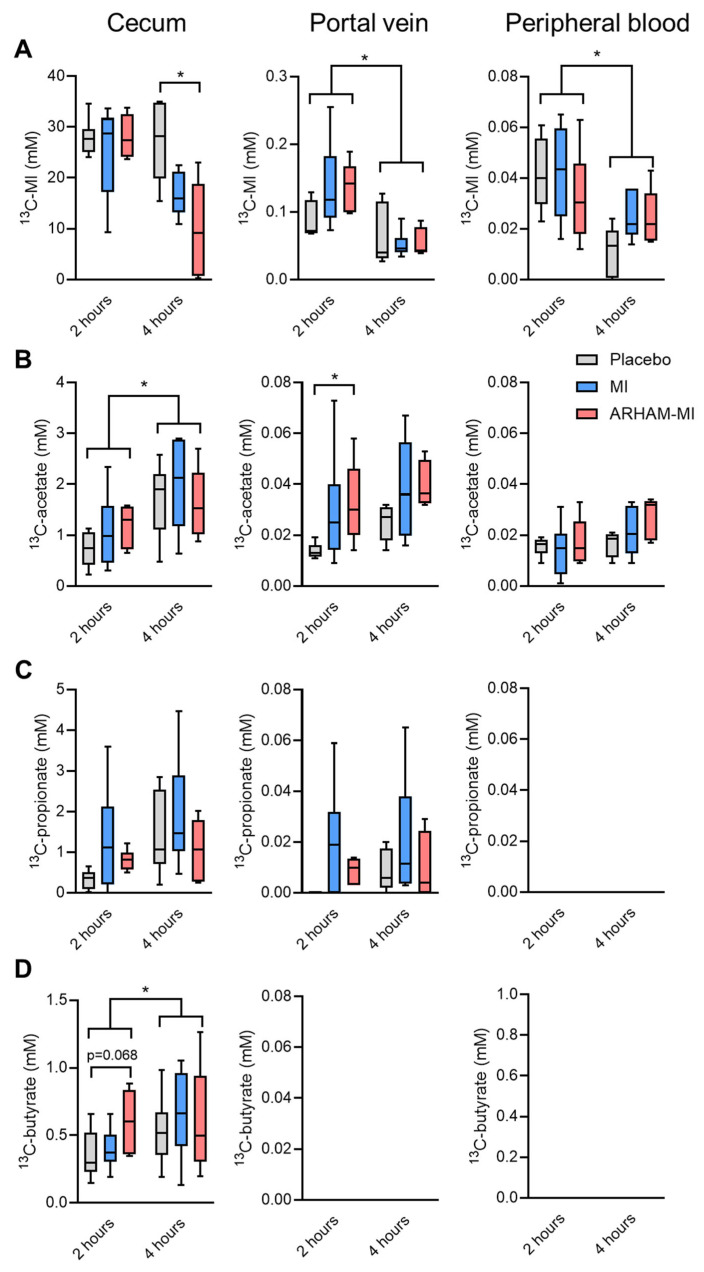
Concentrations of (**A**) ^13^C_6_-MI and its derived SCFAs ^13^C-acetate (**B**), ^13^C-propionate (**C**), and ^13^C-butyrate (**D**) in the cecum, portal vein, and the peripheral circulation of HFD mice treated for 13 weeks with ARHAM-MI or MI alone. * *p* < 0.05 in 2-way ANOVA or 1-way ANOVA compared to ARHAM-MI. Data shown as boxplots with 25th–75th percentiles and whiskers following Tukey’s method, *n* = 5–6 mice per group.

**Figure 4 ijms-26-09340-f004:**
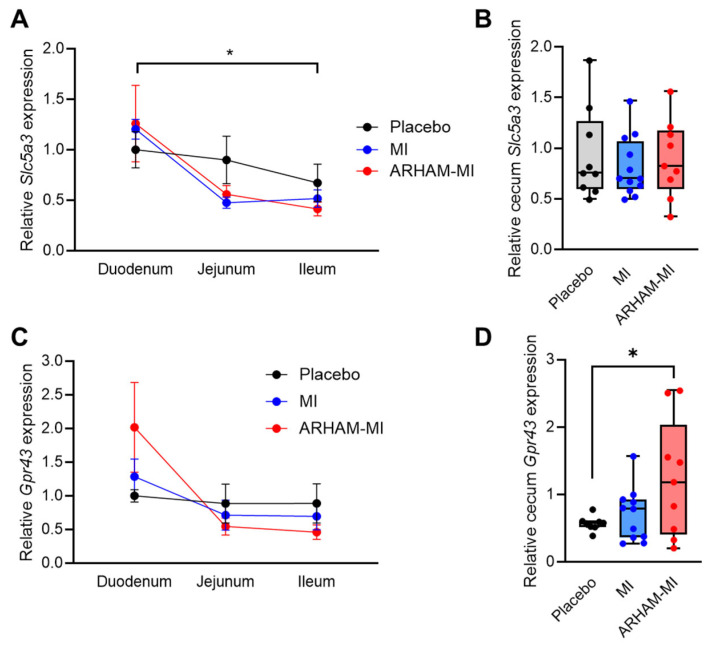
Differences in mRNA expression of the genes encoding the MI transporter Slc5a3 (**A**,**B**) and SCFA receptor Gpr43 (**C**,**D**) in the duodenum, ileum, jejunum (**A**,**C**), and cecum (**B**,**D**) of HFD mice treated for 13 weeks with ARHAM-MI or MI alone. Values are relative to housekeeping genes *Rn18s* and *36b4* and to the expression in the duodenum (**A**,**C**) or the cecum (**B**,**D**) of placebo-treated mice. * *p* < 0.05 in 2-way ANOVA or 1-way ANOVA compared to ARHAM-MI. Data shown as averages ± SEM or boxplots with 25th–75th percentiles and whiskers following Tukey’s method, *n* = 8–12 mice per group.

**Figure 5 ijms-26-09340-f005:**
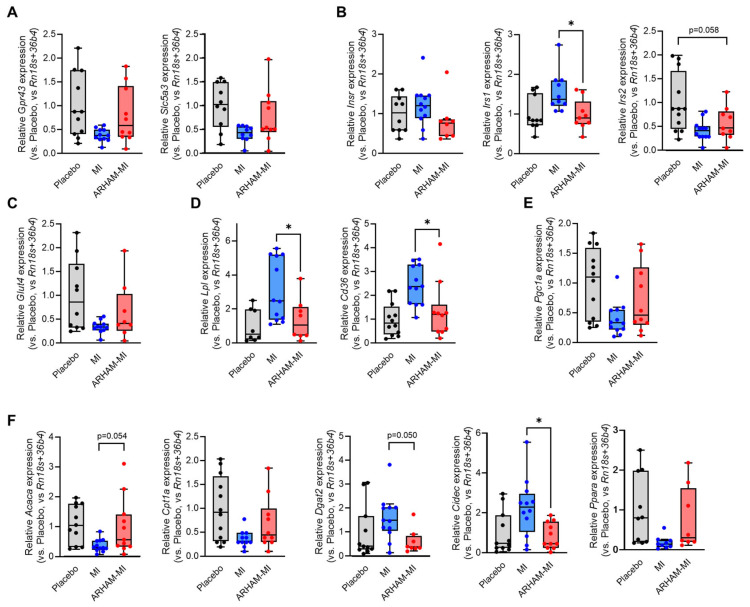
Differences in iWAT mRNA expression of the genes encoding the MI transporter Slc5a3 and the SCFA receptor Gpr43 (**A**), the insulin receptor (Insr) and its downstream substrates Irs1 and Irs2 (**B**), the glucose transporter-4 (**C**), proteins involved in cellular lipid uptake (**D**), the browning-associated protein Pgc1a (**E**), and the proteins involved in cellular lipid synthesis and oxidation (**F**) of HFD mice treated for 13 weeks with ARHAM-MI or MI alone. * *p* < 0.05 in 1-way ANOVA compared to ARHAM-MI. Data shown as averages ± SEM or boxplots with 25th–75th percentiles and whiskers following Tukey’s method, *n* = 8–12 mice per group.

## Data Availability

The original contributions presented in this study are included in the article/Appendix A. Further inquiries can be directed to the corresponding author.

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
