# Peer review of "Intestinal Myo-Inositol Metabolism and Metabolic Effects of Myo-Inositol Utilizing Anaerostipes rhamnosivorans in Mice"

_ijms, 2025, doi:10.3390/ijms26199340_

Round 1
Reviewer 1 Report
Comments and Suggestions for Authors
This manuscript describes the investigation on the effects of a 13-week ARHAM-MI supplementation in high-fat diet-fed mice and the examination of the metabolic fate of MI, including its microbial conversion into short-chain fatty acids (SCFAs), using 13C-MI and stable isotope tracers in the cecum, portal vein, and peripheral blood. The results showed that ARHAM-MI co-supplementation confers modest metabolic benefits in high-fat diet-fed mice.
A few concerns:
- The short-chain fatty acids (SCFAs) used in this manuscript-acetate, propionate and butyrate-seem not contribute to the benefits of the treatment. Maybe the authors should explore other components in the treatment?
- It would be helpful to dose the mice with ARHAM alone.
- On page 10, line 309-311: “Notably, our approach employed low-dose, intermittent oral administration of MI, in contrast to most studies that use continuous dietary supplementation.” What is the rationale for the low-dose intermittent oral administration of MI?
Author Response
Comment 1: The short-chain fatty acids (SCFAs) used in this manuscript -acetate, propionate and butyrate- seem not contribute to the benefits of the treatment. Maybe the authors should explore other components in the treatment?
Answer: The reviewer is right that other components of the treatment, e.g., microbial-derived metabolites in the circulation and a direct microbial effect on the endothelial barrier function, might also have had an impact. We, however, focused on SCFAs because previous research showed that ARHAM can convert MI to propionate and that reduced fasting plasma glucose concentrations upon ARHAM treatment of mice were associated with elevated fecal propionate concentrations. A large part of the data in our manuscript primarily focusses on the fermentation of MI into SCFAs such as propionate. The data in Figure 3 clearly show that 13C-MI is quickly converted into 13C-propionate and that concentrations of 13C-acetate and 13C-butyrate increase over time after the 13C-MI bolus which may contribute to enhanced expression of Gpr43 receptor (Figure 4D) in mouse cecum. Altogether, this underscores, in our opinion, the relevance of studying the role of SCFAs in the MI-ARHAM co-treatment. Future research might however be aimed at exploring other microbiome components. The revised version of our manuscript now addresses this on page 10 (lines 305-309): The current studies mainly focused on the role of the SCFAs in the effects of the AR-HAM-MI co-treatment. It should however be noted that other microbial-derived metabolites as well as direct effects of the microbiome and its metabolites on the epithelial barrier function might also have had an impact. Future research might be directed to study the role of these factors in controlling metabolic health.
Comment 2: It would be helpful to dose the mice with ARHAM alone.
Answer: One of the primary research questions of our studies is whether microbiome-mediated conversion of MI would be beneficial for metabolic health of HFD mice. For this, we not only treated mice with MI but also have a group of mice given ARHAM together with MI to ensure MI conversion. Since the research is not aimed to explore the effects of ARHAM supplementation alone, we did not include a group of mice that were only given ARHAM.
Comment 3: On page 10, line 309-311: “Notably, our approach employed low-dose, intermittent oral administration of MI, in contrast to most studies that use continuous dietary supplementation.” What is the rationale for the low-dose intermittent oral administration of MI?
Answer: Our studies included a group of mice that were given ARHAM together with MI. We had to use gavages to give the bacteria and therefore decided to supply MI by gavage as well to allow comparison between the groups of mice. This resulted in an intermittent administration of a relatively low-dose of MI while other studies supplemented MI in the diets.
Reviewer 2 Report
Comments and Suggestions for Authors
The MS “ Intestinal Myo-Inositol metabolism and Metabolic Effects of 2 myo-Inositol Utilizing Anaerostipes rhamnosivorans in Mice “ is devoted to the study of the effect of ARHAM-MI supplementation in high-fat diet-fed mice and the establishment of the MI conversion pathway.
Overall, the paper is of interest to readers and corresponds to the profile of the IJMS journal. However, the manuscript requires some revision.
Fig. 2. “AUG ITT” and "AUC OGTT" - please provide a description as the Material and Method section is at the end. No matches found throughout the document.
In the figure S1 in the legend, please explain what BW means
Author Response
Comment 1: Fig. 2. “AUG ITT” and "AUC OGTT" - please provide a description as the Material and Method section is at the end. No matches found throughout the document.
Answer: We apologize for this omission and have added the following sentence at the end of the section that describe the ITT and the one for the OGTT (page 11, lines 349-350, and lines 356-357): The area under the curve (AUC) of the blood glucose vs. time curve was calculated using GraphPad Prism 10.
Comment 2: In the figure S1 in the legend, please explain what BW means.
Answer: We thank the reviewer for their meticulous review. We indeed forgot to add this information to the figure legend. The revised version of the S1 figure legend contains an explanation for BW.
Round 2
Reviewer 1 Report
Comments and Suggestions for Authors
While the authors addressed most of my concerns with this resubmitted manuscript. I still think it is worthwhile to dose the mice with ARHAM alone. Although the authors claimed that the research is not aimed to explore the effects of ARHAM supplementation alone, it provides foundation for all the other experiments.
Author Response
Comment: While the authors addressed most of my concerns with this resubmitted manuscript. I still think it is worthwhile to dose the mice with ARHAM alone. Although the authors claimed that the research is not aimed to explore the effects of ARHAM supplementation alone, it provides foundation for all the other experiments.
Answer: The authors agree with the reviewer that including an additional arm with ARHAM alone would have made the study more comprehensive. However, we addressed this question in our earlier work published in Nature Communication 2021 (https://www.nature.com/articles/s41467-021-25081-w#Sec2), where we demonstrated that ARHAM supplementation by ifself had only minor effects, and improvements in glucose handling were observed only when myo-inositol was co-supplemented. Based on those findings, we decided not to include ARHAM-only arm in the present study, as it would not alter the overall conclusion that the active conversion of MI by ARHAM underlies the observed metabolic benefits.